# LEARNING WITH FEATURE-DEPENDENT LABEL NOISE: A PROGRESSIVE APPROACH

**Yikai Zhang**[1][*], **Songzhu Zheng**[2][*], **Pengxiang Wu**[1][*], **Mayank Goswami**[3], **Chao Chen**[2]
[1]Rutgers University, {`yz422,pw241`}`@cs.rutgers.edu`
[2]Stony Brook University, {`zheng.songzhu,chao.chen.1`}`@stonybrook.edu`
[3]City University of New York, `mayank.isi@gmail.com`

## ABSTRACT

Label noise is frequently observed in real-world large-scale datasets. The noise is introduced due to a variety of reasons; it is heterogeneous and feature-dependent. Most existing approaches to handling noisy labels fall into two categories: they either assume an ideal feature-independent noise, or remain heuristic without theoretical guarantees. In this paper, we propose to target a new family of feature-dependent label noise, which is much more general than commonly used i.i.d. label noise and encompasses a broad spectrum of noise patterns. Focusing on this general noise family, we propose a progressive label correction algorithm that iteratively corrects labels and refines the model. We provide theoretical guarantees showing that for a wide variety of (unknown) noise patterns, a classifier trained with this strategy converges to be consistent with the Bayes classifier. In experiments, our method outperforms SOTA baselines and is robust to various noise types and levels.

## 1 INTRODUCTION

Addressing noise in training set labels is an important problem in supervised learning. Incorrect annotation of data is inevitable in large-scale data collection, due to intrinsic ambiguity of data/class and mistakes of human/automatic annotators (Yan et al., 2014; Andreas et al., 2017). Developing methods that are resilient to label noise is therefore crucial in real-life applications.

Classical approaches take a rather simplistic *i.i.d. assumption* on the label noise, i.e., the label corruption is independent and identically distributed and thus is feature-independent. Methods based on this assumption either explicitly estimate the noise pattern (Reed et al., 2014; Patrini et al., 2017; Dan et al., 2019; Xu et al., 2019) or introduce extra regularizer/loss terms (Natarajan et al., 2013; Van Rooyen et al., 2015; Xiao et al., 2015; Zhang & Sabuncu, 2018; Ma et al., 2018; Arazo et al., 2019; Shen & Sanghavi, 2019). Some results prove that the commonly used losses are naturally robust against such i.i.d. label noise (Manwani & Sastry, 2013; Ghosh et al., 2015; Gao et al., 2016; Ghosh et al., 2017; Charoenphakdee et al., 2019; Hu et al., 2020).

Although these methods come with theoretical guarantees, they usually do not perform as well as expected in practice due to the unrealistic i.i.d. assumption on noise. This is likely because *label noise is heterogeneous and feature-dependent.* A cat with an intrinsically ambiguous appearance is more likely to be mislabeled as a dog. An image with poor lighting or severe occlusion can be mislabeled, as important visual clues are imperceptible. Methods that can combat label noise of a much more general form are very much needed to address real-world challenges.

To adapt to the heterogeneous label noise, state-of-the-arts (SOTAs) often resort to a *data-recalibrating* strategy. They progressively identify trustworthy data or correct data labels, and then train using these data (Tanaka et al., 2018; Wang et al., 2018; Lu et al., 2018; Li et al., 2019). The models gradually improve as more clean data are collected or more labels are corrected, eventually converging to models of high accuracy. These data-recalibrating methods best leverage the learning power of deep neural nets and achieve superior performance in practice. However, their underlying mechanism remains a mystery. No methods in this category can provide theoretical insights as to why the model

---

[*]Equal contributions.

can converge to an ideal one. Thus, these methods require careful hyperparameter tuning and are hard to generalize.

In this paper, we propose a novel and principled method that specifically targets the heterogeneous, feature-dependent label noise. Unlike previous methods, we target a much more general family of noise, called *Polynomial Margin Diminishing* (PMD) label noise. In this noise family, we allow arbitrary noise level except for data far away from the true decision boundary. This is consistent with the real-world scenario; data near the decision boundary are harder to distinguish and more likely to be mislabeled. Meanwhile, a datum far away from the decision boundary is a typical example of its true class and should have a reasonably bounded noise level.

Assuming this new PMD noise family, we propose a theoretically-guaranteed data-recalibrating algorithm that gradually corrects labels based on the noisy classifier's confidence. We start from data points with high confidence, and correct the labels of these data using the predictions of the noisy classifier. Next, the model is improved using cleaned labels. We continue alternating the label correction and model improvement until it converges. See Figure 1 for an illustration. Our main theorem shows that with a theory-informed criterion for label correction at each iteration, the improvement of the label purity is guaranteed. Thus the model is guaranteed to improve with sufficient rate through iterations and eventually becomes consistent with the Bayes optimal classifier.

Beside the theoretical strength, we also demonstrate the power of our method in practice. Our method outperforms others on CIFAR-10/100 with various synthetic noise patterns. We also evaluate our method against SOTAs on three real-world datasets with unknown noise patterns.

To the best of our knowledge, our method is the first data-recalibrating method that is theoretically guaranteed to converge to an ideal model. The PMD noise family encompasses a broad spectrum of heterogeneous and feature-dependent noise, and better approximates the real-world scenario. It also provides a novel theoretical setting for the study of label noise.

**Related works.** We review works that do not assume an i.i.d. label noise. Menon et al. (2018) generalized the work of (Ghosh et al., 2015) and provided an elegant theoretical framework, showing that loss functions fulfilling certain conditions naturally resist instance-dependent noise. The method can achieve even better theoretical properties (i.e., Bayes-consistency) with stronger assumption on the clean posterior probability $\eta$. In practice, this method has not been extended to deep neural networks. Cheng et al. (2020) proposed an active learning method for instance-dependent label noise.

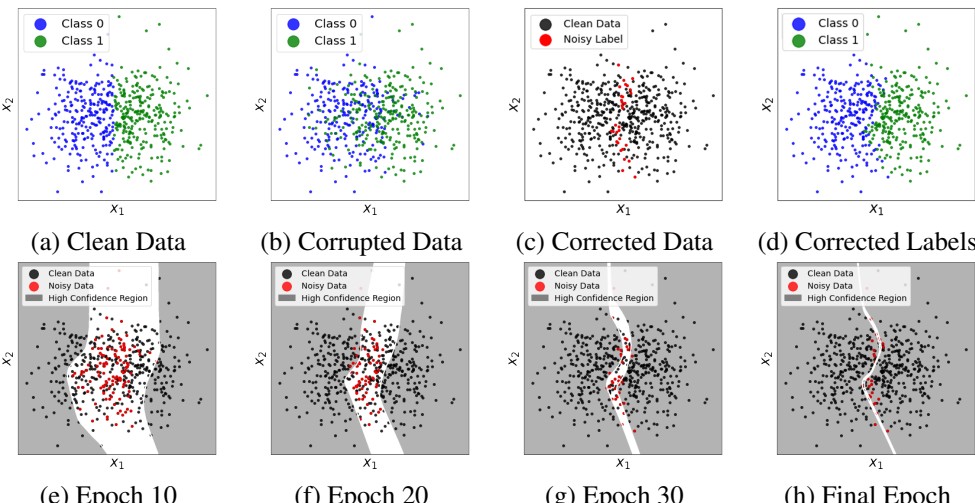

Figure 1: Illustration of the algorithm using synthetic data. (a) Gaussian blob with clean label ($\eta^*(\boldsymbol{x})$). (b) Data with corrupted labels. (c) Final corrected data. Black dots are the data that have their clean labels. Red dots are the noisy data. Points that remain un-corrected are closer to the decision boundary. Our algorithm corrects most of the noise only using noisy classifier's confidence. (d) Data after label correction. (e)-(h) We show the intermediate results at different iterations. Gray region is the area where the classifier has high confidence. Labels within this region are corrected.

The algorithm iteratively queries clean labels from an oracle on carefully selected data. However, this approach is not applicable to settings where kosher annotations are unavailable. Another contemporary work (Chen et al., 2021) showed that the noise in real-world dataset is unlikely to be i.i.d., and proposed to fix the noisy labels by averaging the network predictions on each instance over the whole training process. While being effective, their method lacks theoretical guarantees. Chen et al. (2019) showed by regulating the topology of a classifier's decision boundary, one can improve the model's robustness against label noise.

Data-recalibrating methods use noisy networks' predictions to iteratively select/correct data and improve the models. Tanaka et al. (2018) introduced a joint training framework which simultaneously enforces the network to be consistent with its own predictions and corrects the noisy labels during training. Wang et al. (2018) identified noisy labels as outliers based on their label consistencies with surrounding data. Lu et al. (2018) used a curriculum learning strategy where the teacher net is trained on a small kosher dataset to determine if a datum is clean; then the learnt curriculum that gives the weight to each datum is fed into the student net for the training and inference. (Yu et al., 2019; Bo et al., 2018) trained two synchronized networks; the confidence and consistency of the two networks are utilized to identify clean data. Wu et al. (2020) selected the clean data by investigating the topological structures of the training data in the learned feature space. For completeness, we also refer to other methods of similar design (Li et al., 2017; Vahdat, 2017; Andreas et al., 2017; Ma et al., 2018; Thulasidasan et al., 2019; Arazo et al., 2019; Shu et al., 2019; Yi & Wu, 2019).

As for theoretical guarantees, Ren et al. (2018) proposed an algorithm that iteratively re-weights each data point by solving an optimization problem. They proved the convergence of the training, but provided no guarantees that the model converges to an ideal one. Amid et al. (2019b) generalized the work of (Amid et al., 2019a) and proposed a tempered matching loss. They showed that when the final softmax layer is replaced by the bi-tempered loss, the resulting classifier will be Bayes consistent. Zheng et al. (2020) proved a one-shot guarantee for their data-recalibrating method; but the convergence of the model is not guaranteed. *Our method is the first data-recalibrating method which is guaranteed to converge to a well-behaved classifier.*

## 2 METHOD

We start by introducing the family of Poly-Margin Diminishing (PMD) label noise. In Section 2.2, we present our main algorithm. Finally, we prove the correctness of our algorithm in Section 3.

**Notations and preliminaries.** Although the noise setting and algorithm naturally generalize to multi-class, for simplicity we focus on binary classification. Let the feature space be $\mathcal{X}$. We assume the data $(\boldsymbol{x}, y)$ is sampled from an underlying distribution $D$ on $\mathcal{X} \times \{0, 1\}$. Define the posterior probability $\eta(\boldsymbol{x}) = \mathbb{P}[y = 1 \mid \boldsymbol{x}]$. Let $\tau_{0,1}(\boldsymbol{x}) = \mathbb{P}[\widetilde{y} = 1 \mid y = 0, \boldsymbol{x}]$ and $\tau_{1,0}(\boldsymbol{x}) = \mathbb{P}[\widetilde{y} = 0 \mid y = 1, \boldsymbol{x}]$ be the noise functions, where $\widetilde{y}$ denotes the corrupted label. For example, if a datum $\boldsymbol{x}$ has true label $y = 0$, it has $\tau_{0,1}(\boldsymbol{x})$ chance to be corrupted to 1. Similarly, it has $\tau_{1,0}(\boldsymbol{x})$ chance to be corrupted from 1 to 0. Let $\widetilde{\eta}(\boldsymbol{x}) = \mathbb{P}[\widetilde{y} = 1 \mid \boldsymbol{x}]$ be the noisy posterior probability of $\widetilde{y} = 1$ given feature $\boldsymbol{x}$. Let $\eta^*(\boldsymbol{x}) = \mathbb{I}_{\{\eta(\boldsymbol{x}) \geq \frac{1}{2}\}}$ be the (clean) Bayes optimal classifier, where $\mathbb{I}_A$ equals 1 if $A$ is true, and 0 otherwise. Finally, let $f(\boldsymbol{x}) : \mathcal{X} \to [0, 1]$ be the classifier scoring function (the softmax output of a neural network in this paper).

### 2.1 POLY-MARGIN DIMINISHING NOISE

We first introduce the family of noise functions $\tau$ this paper will address. We introduce the concept of *polynomial margin diminishing noise* (PMD noise), which only upper bounds the noise $\tau$ in a certain level set of $\eta(\boldsymbol{x})$, thus allowing $\tau$ to be arbitrarily high outside the restricted domain. This formulation not only covers the feature-independent scenario but also generalizes scenarios proposed by (Du & Cai, 2015; Menon et al., 2018; Cheng et al., 2020).

**Definition 1** (PMD noise). *A pair of noise functions $\tau_{0,1}(\boldsymbol{x})$ and $\tau_{1,0}(\boldsymbol{x})$ are polynomial-margin diminishing (PMD), if there exist constants $t_0 \in (0, \frac{1}{2})$, and $c_1, c_2 > 0$ such that:*

$$\tau_{1,0}(\boldsymbol{x}) \leq c_1[1 - \eta(\boldsymbol{x})]^{1+c_2}; \forall \eta(\boldsymbol{x}) \geq \frac{1}{2} + t_0, \text{ and}$$

$$\tau_{0,1}(\boldsymbol{x}) \leq c_1\eta(\boldsymbol{x})^{1+c_2}; \forall \eta(\boldsymbol{x}) \leq \frac{1}{2} - t_0. \tag{1}$$

We abuse notation by referring to $t_0$ as the "margin" of $\tau$. Note that the PMD condition only requires the *upper bound* on $\tau$ to be polynomial and monotonically decreasing in the region where the Bayes classifier is fairly confident. For the region $\{\boldsymbol{x} : |\eta(\boldsymbol{x}) - \frac{1}{2}| < t_0\}$, we allow both $\tau_{0,1}(\boldsymbol{x})$ and $\tau_{1,0}(\boldsymbol{x})$ to be arbitrary. Figure 2(d) illustrates the upper bound (orange curve) and a sample noise function (blue curve). We also show the corrupted data according to this noise function (black points are the clean data whereas red points are the data with corrupted labels).

The PMD noise family is much more general than existing noise assumptions. For example, the boundary consistent noise (BCN) (Du & Cai, 2015; Menon et al., 2018) assumes a noise function that monotonically decreases as the data are moving away from the decision boundary. See Figure 2(c) for an illustration. This noise is much more restrictive compared to our PMD noise which (1) only requires a monotonic upper bound, and (2) allows arbitrary noise strength in a wide buffer near the decision boundary. Figure 2(b) shows a traditional feature-independent noise pattern (Reed et al., 2014; Patrini et al., 2017), which assumes $\tau_{0,1}(\boldsymbol{x})$ (resp. $\tau_{1,0}(\boldsymbol{x})$) to be a constant independent of $\boldsymbol{x}$.

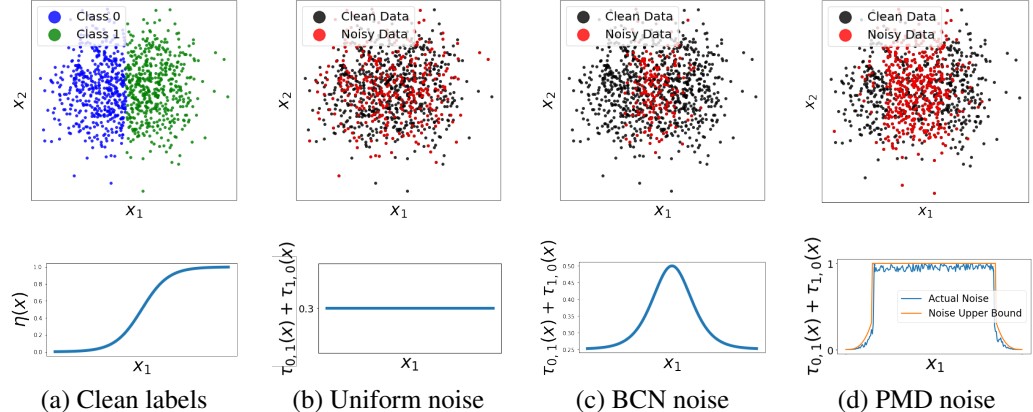

(a) Clean labels  (b) Uniform noise  (c) BCN noise  (d) PMD noise

Figure 2: Illustration of different noise functions. (a) The original data: Gaussian blob with clean labels (by clean label, we refer to the prediction of the Bayes optimal classifier $\eta^*(\boldsymbol{x})$, not $y$). Confident region of $\eta$ (and thus $f$) in this case is the place where $\eta(\boldsymbol{x})$ is close to 0 or 1. Blue and green dots correspond to different classes. (b) Uniform label noise: each point has an equal probability to be flipped. Red dots are data with corrupted labels; black dots correspond to data that are not corrupted. (c) BCN noise: the level of noise is decreasing as $\eta^*(\boldsymbol{x})$ becomes confident. (d) PMD noise: noise level (blue) is only upper bounded by diminishing polynomial function when $\eta(\boldsymbol{x})$ is higher or lower than certain threshold. The upper bound is shown in solid orange curve. The dashed orange curve means the noise level near the decision boundary is unbounded.

## 2.2 The Progressive Correction Algorithm

Our algorithm iteratively trains a neural network and corrects labels. We start with a warm-up period, in which we train the neural network (NN) with the original noisy data. This allows us to attain a reasonable network before it starts fitting noise (Zhang et al., 2017). After the warm-up period, the classifier can be used for label correction. We only correct a label on which the classifier $f$ has a very high confidence. The intuition is that under the noise assumption, there exists a "pure region" in which the prediction of the noisy classifier $f$ is highly confident and is consistent with the clean Bayes optimal classifier $\eta^*$. Thus the label correction gives clean labels within this pure region. In particular, we select a high threshold $\theta$. If $f$ predicts a different label than $\tilde{y}$ and its confidence is above the threshold, $|f(\boldsymbol{x}) - 1/2| > \theta$, we flip the label $\tilde{y}$ to the prediction of $f$. We repeatedly correct labels and improve the network until no label is corrected. Next, we slightly decrease the threshold $\theta$, use the decreased threshold for label correction, and improve the model accordingly. We continue the process until convergence. For convenience in theoretical analysis, in the algorithm, we define a continuous increasing threshold $T$ and let $\theta = 1/2 - T$. Our algorithm is summarized in Algorithm 1. We term our algorithm as PLC (Progressive Label Correction). In Section 3, we will show that this iterative algorithm will converge to be consistent with clean Bayes optimal classifier $\eta^*(\boldsymbol{x})$ for most of the input instances.

---

**Algorithm 1** `Progressive Label Correction`

---

**Input:** Dataset $\tilde{S} = \{(\boldsymbol{x}_1, \widetilde{y}_1^0), \cdots, (\boldsymbol{x}_n, \widetilde{y}_n^0)\}$, initial NN $f(\boldsymbol{x})$, step size $\beta$,
  initial and end thresholds $(T_0, T_{end})$, warm-up $m$, total round $N$
**Output:** $f_{final}(\cdot)$
 1: $T \leftarrow T_0$
 2: $\theta \leftarrow 1/2 - T_0$
 3: **for** $t \leftarrow 1, \cdots, N$ **do**
 4:    Train $f(\boldsymbol{x})$ on $\tilde{S}$
 5:    **for all** $(\boldsymbol{x}_i, \widetilde{y}_i^{t-1}) \in \tilde{S}$ **and** $|f(\boldsymbol{x}_i) - \frac{1}{2}| \geq \theta$ **do**
 6:        $\widetilde{y}_i^t \leftarrow \mathbb{I}_{\{f(\boldsymbol{x}_i) \geq \frac{1}{2}\}}$
 7:    **end for**
 8:    **if** $t \geq m$ **then**
 9:        $\theta \leftarrow 1/2 - T$
10:       **if** $\forall i \in [1, \cdots, n], \widetilde{y}_i^t = \widetilde{y}_i^{t-1}$ **then**
11:           $T \leftarrow \min(T(1 + \beta), T_{end})$
12:       **end if**
13:    **end if**
14:    $\tilde{S} \leftarrow \{(\boldsymbol{x}_1, \widetilde{y}_1^t), \cdots, (\boldsymbol{x}_n, \widetilde{y}_n^t)\}$
15: **end for**

---

**Generalizing to the multi-class scenario.** In multi-class scenario, denote by $f_i(\boldsymbol{x})$ the classifier's prediction probability of label $i$. Let $h_{\boldsymbol{x}}$ be the classifier's class prediction, i.e., $h_{\boldsymbol{x}} = \arg\max_i f_i(\boldsymbol{x})$. We change the $|f(\boldsymbol{x}) - \frac{1}{2}|$ term to the gap between the highest confidence $f_{h_{\boldsymbol{x}}}(\boldsymbol{x})$ and the confidence on $\widetilde{y}$, $f_{\widetilde{y}}(\boldsymbol{x})$. If the absolute difference between these two confidences is larger than certain threshold $\theta$, then we correct $\widetilde{y}$ to $h_{\boldsymbol{x}}$. In practice, we find using the difference of logarithms will be more robust.

## 3 ANALYSIS

Our analysis focuses on the asymptotic case and answers the following question: Given infinitely many data with corrupted labels, is it possible to learn a reasonably good classifier? We show that if the noise satisfies the arguably-general PMD condition, the answer is yes. Assuming mild conditions on the hypothesis class of the machine learning model and the distribution $D$, we prove that Algorithm 1 obtains a nearly clean classifier. This reduces the challenge of noisy label learning from a realizable problem into a sample complexity problem. In this work we only focus on the asymptotic case, and leave the sample complexity for future work.

### 3.1 ASSUMPTIONS

Our first assumption restricts the model to be able to at least approximate the true Bayes classifier $\eta(\boldsymbol{x})$. This condition assumes that given a hypothesis class $\mathcal{H}$ with sufficient complexity, the approximation gap between a classifier $f(\boldsymbol{x})$ in this class and $\eta(\boldsymbol{x})$ is determined by the inconsistency between the noisy labels and the Bayes optimal classifier.

**Definition 2** (Level set $(\alpha, \epsilon)$-consistency)**.** *Suppose data are sampled as $(\boldsymbol{x}, \tilde{y}) \sim D(\boldsymbol{x}, \tilde{\eta}(\boldsymbol{x}))$ and $f(\boldsymbol{x}) = \arg\min_{h \in \mathcal{H}} E_{(\boldsymbol{x}, \tilde{y}) \sim D(\boldsymbol{x}, \tilde{\eta}(\boldsymbol{x}))} Loss(h(\boldsymbol{x}), \tilde{y})$. Given $\varepsilon < \frac{1}{2}$, we call $\mathcal{H}$ is $(\alpha, \epsilon)$-consistent if:*

$$|f(\boldsymbol{x}) - \eta(\boldsymbol{x})| \leq \alpha \mathbb{E}_{(\boldsymbol{z}, \tilde{y}) \sim D(\boldsymbol{z}, \tilde{\eta}(\boldsymbol{z}))} \left[ \mathbb{1}_{\{\tilde{y}_{\boldsymbol{z}} \neq \eta^*(\boldsymbol{z})\}}(\boldsymbol{z}) \,\middle|\, \left|\eta(\boldsymbol{z}) - \frac{1}{2}\right| \geq \left|\eta(\boldsymbol{x}) - \frac{1}{2}\right| \right] + \epsilon. \quad (2)$$

For two input instances $\boldsymbol{z}$ and $\boldsymbol{x}$ such that $\eta(\boldsymbol{z}) > \eta(\boldsymbol{x})$ (and hence the clean Bayes optimal classifier $\eta^*(\boldsymbol{x})$ has higher confidence at $\boldsymbol{z}$ than it does at $\boldsymbol{x}$), the indicator function $\mathbb{1}_{\{\tilde{y}_{\boldsymbol{z}} \neq \eta^*(\boldsymbol{z})\}} \left(\boldsymbol{z} : \left|\eta(\boldsymbol{z}) - \frac{1}{2}\right| \geq \left|\eta(\boldsymbol{x}) - \frac{1}{2}\right|\right)$ equals to 1 if the label of the more confident point $\boldsymbol{z}$ is inconsistent with $\eta^*(\boldsymbol{x})$. This condition says that the approximation error of the classifier at $\boldsymbol{x}$ should be controlled by the risk of $\eta^*(\cdot)$ at points $\boldsymbol{z}$ where $\eta^*(\cdot)$ is more confident than it is at $\boldsymbol{x}$.

We next define a regularity condition of data distribution which describes the continuity of the level set density function.

**Definition 3** (Level set bounded distribution). *Define the margin $t(\boldsymbol{x}) = |\eta(\boldsymbol{x}) - \frac{1}{2}|$ and $G(t)$ be the cdf of $t$: $G(t) = \mathbb{P}_{\boldsymbol{x} \sim D}(|\eta(\boldsymbol{x}) - \frac{1}{2}| \leq t)$. Let $g(t) = G'(t)$ be the density function of $t$. We say the distribution $D$ is $(c_*, c^*)$-bounded if for all $0 \leq t \leq 1/2$, $0 < c_* \leq g(t) \leq c^*$. If $D$ is $(c_*, c^*)$-bounded, we define the worst-case density-imbalance ratio of $D$ by $\ell_D := \frac{c^*}{c_*}$.*

The above condition enforces the continuity of level set density function. This is crucial in the analysis since such a continuity allows one to borrow information from its neighborhood region so that a clean neighbor can help correct the corrupted label. To simplify the notation, we will omit $D$ in the subscript when we mention $\ell$. From now on, we will assume:

**Assumption 1.** There exist constants $\alpha, \epsilon, c_*, c^*$ such that the hypothesis class $\mathcal{H}$ is $(\alpha, \epsilon)$-consistent and the unknown distribution $D$ is $(c_*, c^*)$-bounded.

## 3.2 MAIN RESULT AND PROOF SKETCH

In this section we first state our main result, and then present the supporting claims. Complete proofs can be found in the appendix. Our main result below states that if our starting function is trained correctly, i.e., $f(\boldsymbol{x}) = \arg\min_{h \in \mathcal{H}} E_{(\boldsymbol{x}, \tilde{y}) \sim D(\boldsymbol{x}, \tilde{\eta}(\boldsymbol{x}))} Loss(h(\boldsymbol{x}), \tilde{y})$, then Algorithm 1 terminates with most of the final labels matching the Bayes optimal classifier labels. In practice, minimizing true risk is not achievable. Instead, the empirical risk is used to estimate true risk, approaching true risk asymptotically. For a scoring function $f$, we will denote by $y_{f(\boldsymbol{x})} := \mathbb{I}(f(\boldsymbol{x}) \geq 1/2)$ the label predicted by $f$.

**Theorem 1.** *Under Assumption 1, for any noise $\tau$ which is PMD with margin $t_0$, define $e_0 = max(t_0, \frac{\alpha + \varepsilon}{1 + 2\alpha})$. Then for the output of Algorithm 1 with $f$ as above and with the following initializations: (1) $T_0 < \frac{1}{2} - e_0$, (2) $m \geq \frac{\ell \alpha}{\varepsilon} \log(\frac{2T_0}{1 - 2e_0})$, (3) $N \geq m + \frac{1}{\beta} \log(\frac{T_0}{3\varepsilon})$, (4) $T_{end} \leq 3\epsilon$ and (5) $\frac{\varepsilon}{\alpha \ell} \leq \beta \leq \frac{2\varepsilon}{\alpha \ell}$, we have:*
$$\mathbb{P}_{\boldsymbol{x} \sim D}[y_{f_{final}(\boldsymbol{x})} = \eta^*(\boldsymbol{x})] \geq 1 - 3c^*\epsilon.$$

In the remainder of this section we shall assume that the noise $\tau$ is PMD with margin $t_0$. To prove our result we first define a "pure" level set.

**Definition 4** (Pure $(e, f, \eta)$-level set). *A set $L(e, \eta) := \{\boldsymbol{x} \mid |\eta(\boldsymbol{x}) - \frac{1}{2}| \geq e\}$ is pure for $f$ if $y_{f(\boldsymbol{x})} = \eta^*(\boldsymbol{x})$ for all $\boldsymbol{x} \in L(e, \eta)$.*

We now state a lemma that forms the foundation of our progressive correction algorithm. We show that given a tiny region where the model is reliable, we can move one step forward by trusting the model. Although the improvement is slight in a single round, it empowers a conservatively recursive step in the Algorithm 1.

**Lemma 1** (One round purity improvement). *Suppose Assumption 1 is satisfied, and assume an $f$ such that there exists a pure $(e, f, \eta)$-level set with $3\epsilon \leq e < \frac{1}{2}$. Let $\tilde{\eta}_{new}(\boldsymbol{x}) = y_{f(\boldsymbol{x})}$ if $|f(\boldsymbol{x}) - 1/2| \geq e$ and $\tilde{\eta}(\boldsymbol{x})$ if $|f(\boldsymbol{x}) - 1/2| < e$, and assume $f_{new} = \arg\min_{h \in \mathcal{H}} E_{(\boldsymbol{x}, \tilde{y}) \sim D(\boldsymbol{x}, \tilde{\eta}_{new}(\boldsymbol{x}))} Loss(h(\boldsymbol{x}), \tilde{y})$. Let $e_{new} = \min\{e \mid e > 0, \ L(e, \eta) \text{ is pure for } f_{new}\}$. Then $\frac{1}{2} - e_{new} \geq (1 + \frac{\varepsilon}{\alpha \ell})(\frac{1}{2} - e)$.*

The above lemma states that the cleansed region will be enlarged by at least a constant factor. In the following lemma, we justify the functionality of the first $m$ warm-up rounds. Since the initial neural network can behave badly, the region where we can trust the classifier can be very limited. Before starting the flipping procedure in a relatively larger level set, one first needs to expand the initial tiny region $\frac{1}{2} - e_0$ to a constant $T_0$.

**Lemma 2** (Warm-up rounds). *Suppose for a given function $f_0$ there exists a level set $L(e_0, \eta)$ which is pure for $f_0$. Given $T_0 < 1/2$, after running Algorithm 1 for $m \geq \frac{\ell \alpha}{\varepsilon} \log(\frac{2T_0}{1 - 2e_0})$ rounds, there exists a level set $L(\frac{1}{2} - T_0, \eta)$ that is pure for $f_0$.*

Next we present our final lemma that combines the previous two lemmata.

**Lemma 3.** *Suppose Assumption 1 is satisfied, and for a given function $f_0$ there exists a level set $L(e_0, \eta)$ which is pure for $f_0$. If one runs Algorithm 1 starting with $f_0$ and the initializations: (1) $T_0 < \frac{1}{2} - e_0$, (2) $m \geq \frac{\ell \alpha}{\varepsilon} \log(\frac{2T_0}{1 - 2e_0})$, (3) $N \geq m + \frac{1}{\beta} \log(\frac{1 - 6\varepsilon}{2T_0})$, (4) $T_{end} \leq \frac{1}{2} - 3\epsilon$ and (5) $\frac{\varepsilon}{\alpha \ell} \leq \beta \leq \frac{2\varepsilon}{\alpha \ell}$, then we have $\mathbb{P}_{\boldsymbol{x} \sim D}[y_{f_{final}(\boldsymbol{x})} = \eta^*(\boldsymbol{x})] \geq 1 - 3c^*\epsilon$.*

This lemma states that given an initial model that has a reasonably pure super level set, one can manage to progressively correct a large fraction of corrupted labels by running Algorithm 1 for a sufficient long time with carefully chosen parameters. The limit of Algorithm 1 will depend on the approximation ability of the neural network, which is characterized by parameter $\varepsilon$ in Definition 2. To prove Theorem 1 using Lemma 3, it suffices to get a model which has a reliable region. This is provably achievable by training with a family of good scoring functions on PMD noisy data.

## 4 EXPERIMENTS

We evaluate our method on both synthetic and real-world datasets. We first conduct synthetic experiments on two public datasets CIFAR-10 and CIFAR-100 (Krizhevsky et al., 2009). To synthesize the label noise, we first approximate the true posterior probability $\eta$ using the confidence prediction of a clean neural network (trained with the original clean labels). We call these original labels *raw labels*. Then we sample $y_x \sim \eta(x)$ for each instance $x$. Instead of using raw labels, we use these sampled labels $y_x$ as the clean labels, whose posterior probabilities are exactly $\eta(x)$; and therefore the neural network is the Bayes optimal classifier $\eta^* : \mathcal{X} \to \{1, \cdots, C\}$, where $C$ is the number of classes. Note that in multi-class setting, $\eta(x)$ has a vector output and $\eta_i(x)$ is the $i$-th element of this vector.

**Noise generation.** We consider a generic family of noise. We consider not only feature-dependent noise, but also hybrid noise that consists of both feature-dependent noise and i.i.d. noise.

For feature-dependent noise, we use three types of noise functions within the PMD noise family. To make the noise challenging enough, for input $x$ we always corrupt label from the most confident category $u_x$ to the second confident category $s_x$, according to $\eta(x)$. Because $s_x$ is the class that confuses $\eta^*(x)$ the most, this noise will hurt the network's performance the most. Note that $y_x$ is sampled from $\eta(x)$, which has quite an extreme confidence. Thus we generally assume $y_x$ is $u_x$. For each datum $x$, we only flip it to $s_x$ or keep it as $u_x$. The three noise functions are as follows:

$$\text{Type-I} : \tau_{u_x,s_x} = -\frac{1}{2}\left[\eta_{u_x}(x) - \eta_{s_x}(x)\right]^2 + \frac{1}{2}, \quad \text{Type-II} : \tau_{u_x,s_x} = 1 - \left[\eta_{u_x}(x) - \eta_{s_x}(x)\right]^3,$$

$$\text{Type-III} : \tau_{u_x,s_x} = 1 - \frac{1}{3}\left[\left[\eta_{u_x}(x) - \eta_{s_x}(x)\right]^3 + \left[\eta_{u_x}(x) - \eta_{s_x}(x)\right]^2 + \left[\eta_{u_x}(x) - \eta_{s_x}(x)\right]\right].$$

Notice that the noise level is determined by the $\eta(x)$ naturally and we cannot control it directly. To change the noise level, we multiply $\tau_{u_x,s_x}$ by a certain constant factor such that the final proportion of noise matches our requirement. For PMD noise only, we test noise levels 35% and 70%, meaning that 35% and 70% of the data are corrupted due to the noise, respectively.

For i.i.d. noise we follow the convention and adopt the commonly used uniform noise and asymmetric noise (Patrini et al., 2017). We artificially corrupt the labels by constructing the noise transition matrix $T$, where $T_{ij} = P(\widetilde{y} = j | y = i) = \tau_{ij}$ defines the probability that a true label $y = i$ is flipped to $j$. Then for each sample with label $i$, we replace its label with the one sampled from the probability distribution given by the $i$-th row of matrix $T$. We consider two kinds of i.i.d. noise in this work. (1) Uniform noise: the true label $i$ is corrupted uniformly to other classes, i.e., $T_{ij} = \tau/(C-1)$ for $i \neq j$, and $T_{ii} = 1 - \tau$, where $\tau$ is the constant noise level; (2) Asymmetric noise: the true label $i$ is flipped to $j$ or stays unchanged with probabilities $T_{ij} = \tau$ and $T_{ii} = 1 - \tau$, respectively.

**Baselines.** We compare our method with several recently proposed approaches. (1) GCE (Zhang & Sabuncu, 2018); (2) Co-teaching+ (Yu et al., 2019); (3) SL (Wang et al., 2019); (4) LRT (Zheng et al., 2020). All these methods are generic and handle the label noise without assuming the noise structures. Finally, we also provide the results by standard method, which simply trains the deep network on noisy datasets in a standard manner.

During training, we use a batch size of 128 and train the network for 180 epochs to ensure the convergence of all methods. We train the network with SGD optimizer, with initial learning rate 0.01. We randomly repeat the experiments 3 times, and report the mean and standard deviation values. Our code is available at `https://github.com/pxiangwu/PLC`.

**Results.** Table 1 lists the performance of different methods under three types of feature-dependent noise at noise levels 35% and 70%. We observe that our method achieves the best performance across different noise settings. Moreover, notice that some of the baseline methods' performances are inferior to the standard approach. Possible reasons are that these methods behave too conservatively

in dealing with noise. Thus they only make use of a small subset of the original training set, which is not representative enough to grant the model good discriminative ability.

In Table 2 we show the results on datasets corrupted with a combination of feature-dependent noise and i.i.d. noise, which ends up to real noise levels ranging from 50% to 70% (in terms of the proportion of corrupted labels). I.i.d. noise is overlayed on the feature-dependent noise. Our method outperforms baselines under these more complicated noise patterns. In contrast, when the noise level is high like the cases where we further apply additional 30% and 60% uniform noise, performances of a few baselines deteriorate and become worse than the standard approach.

We carry out the ablation studies on hyper-parameters $\theta_0$ (determining the initial confidence threshold for label correction, see Algorithm 1) and $\beta$ (the step size). In Tables 3 and 4, we show that our method is robust against the choice of $\theta_0$ and $\beta$ up to a wide range. Notice that to compare against the threshold $\theta_0$, here we are calculating the absolute difference of $\log f_{\widetilde{y}}(\boldsymbol{x})$ and $\log f_{h_{\boldsymbol{x}}}(\boldsymbol{x})$. As mentioned in Section 2.2, this operation gives a good performance in practice.

Table 1: Test accuracy (%) on CIFAR-10 and CIFAR-100 under different feature-dependent noise types and levels. The average accuracy and standard deviation over 3 trials are reported.

| Dataset | Noise | Standard | Co-teaching+ | GCE | SL | LRT | PLC (ours) |
|---------|-------|----------|--------------|-----|-----|-----|------------|
| CIFAR-10 | Type-I ( 35% ) | $78.11 \pm 0.74$ | $79.97 \pm 0.15$ | $80.65 \pm 0.39$ | $79.76 \pm 0.72$ | $80.98 \pm 0.80$ | $\mathbf{82.80 \pm 0.27}$ |
| | Type-I ( 70% ) | $41.98 \pm 1.96$ | $40.69 \pm 1.99$ | $36.52 \pm 1.62$ | $36.29 \pm 0.66$ | $41.52 \pm 4.53$ | $\mathbf{42.74 \pm 2.14}$ |
| | Type-II ( 35% ) | $76.65 \pm 0.57$ | $77.34 \pm 0.44$ | $77.60 \pm 0.88$ | $77.92 \pm 0.89$ | $80.74 \pm 0.25$ | $\mathbf{81.54 \pm 0.47}$ |
| | Type-II ( 70% ) | $45.57 \pm 1.12$ | $45.44 \pm 0.64$ | $40.30 \pm 1.46$ | $41.11 \pm 1.92$ | $44.67 \pm 3.89$ | $\mathbf{46.04 \pm 2.20}$ |
| | Type-III ( 35% ) | $76.89 \pm 0.79$ | $78.38 \pm 0.67$ | $79.18 \pm 0.61$ | $78.81 \pm 0.29$ | $81.08 \pm 0.35$ | $\mathbf{81.50 \pm 0.50}$ |
| | Type-III ( 70% ) | $43.32 \pm 1.00$ | $41.90 \pm 0.86$ | $37.10 \pm 0.59$ | $38.49 \pm 1.46$ | $44.47 \pm 1.23$ | $\mathbf{45.05 \pm 1.13}$ |
| CIFAR-100 | Type-I ( 35% ) | $57.68 \pm 0.29$ | $56.70 \pm 0.71$ | $58.37 \pm 0.18$ | $55.20 \pm 0.33$ | $56.74 \pm 0.34$ | $\mathbf{60.01 \pm 0.43}$ |
| | Type-I ( 70% ) | $39.32 \pm 0.43$ | $39.53 \pm 0.28$ | $40.01 \pm 0.71$ | $40.02 \pm 0.85$ | $45.29 \pm 0.43$ | $\mathbf{45.92 \pm 0.61}$ |
| | Type-II ( 35% ) | $57.83 \pm 0.25$ | $56.57 \pm 0.52$ | $58.11 \pm 1.05$ | $56.10 \pm 0.73$ | $57.25 \pm 0.68$ | $\mathbf{63.68 \pm 0.29}$ |
| | Type-II ( 70% ) | $39.30 \pm 0.32$ | $36.84 \pm 0.39$ | $37.75 \pm 0.46$ | $38.45 \pm 0.45$ | $43.71 \pm 0.51$ | $\mathbf{45.03 \pm 0.50}$ |
| | Type-III ( 35% ) | $56.07 \pm 0.79$ | $55.77 \pm 0.98$ | $57.51 \pm 1.16$ | $56.04 \pm 0.74$ | $56.57 \pm 0.30$ | $\mathbf{63.68 \pm 0.29}$ |
| | Type-III ( 70% ) | $40.01 \pm 0.18$ | $35.37 \pm 2.65$ | $40.53 \pm 0.60$ | $39.94 \pm 0.84$ | $44.41 \pm 0.19$ | $\mathbf{44.45 \pm 0.62}$ |

Table 2: Test accuracy (%) on CIFAR-10 and CIFAR-100 under different hybrid noise types and levels. The average accuracy and standard deviation over 3 trials are reported.

| Dataset | Noise | Standard | Co-teaching+ | GCE | SL | LRT | PLC (ours) |
|---------|-------|----------|--------------|-----|-----|-----|------------|
| CIFAR-10 | Type-I + 30% Uniform | $75.26 \pm 0.32$ | $78.72 \pm 0.53$ | $78.08 \pm 0.66$ | $77.79 \pm 0.46$ | $75.97 \pm 0.27$ | $\mathbf{79.04 \pm 0.50}$ |
| | Type-I + 60% Uniform | $64.25 \pm 0.78$ | $55.49 \pm 2.11$ | $67.43 \pm 1.43$ | $67.63 \pm 1.36$ | $59.22 \pm 0.74$ | $\mathbf{72.21 \pm 2.92}$ |
| | Type-I + 30% Asymmetric | $75.21 \pm 0.64$ | $75.43 \pm 2.96$ | $76.91 \pm 0.56$ | $77.14 \pm 0.70$ | $76.96 \pm 0.45$ | $\mathbf{78.31 \pm 0.41}$ |
| | Type-II + 30% Uniform | $74.92 \pm 0.63$ | $75.19 \pm 0.54$ | $75.69 \pm 0.21$ | $75.08 \pm 0.47$ | $75.94 \pm 0.58$ | $\mathbf{80.08 \pm 0.37}$ |
| | Type-II + 60% Uniform | $64.02 \pm 0.66$ | $59.89 \pm 0.63$ | $66.39 \pm 0.29$ | $66.76 \pm 1.60$ | $58.99 \pm 1.43$ | $\mathbf{71.21 \pm 1.46}$ |
| | Type-II + 30% Asymmetric | $74.28 \pm 0.39$ | $73.37 \pm 0.83$ | $75.30 \pm 0.81$ | $75.43 \pm 0.42$ | $77.03 \pm 0.62$ | $\mathbf{77.63 \pm 0.30}$ |
| | Type-III + 30% Uniform | $74.00 \pm 0.38$ | $77.31 \pm 0.11$ | $77.00 \pm 0.12$ | $76.22 \pm 0.12$ | $75.66 \pm 0.57$ | $\mathbf{80.06 \pm 0.47}$ |
| | Type-III + 60% Uniform | $63.96 \pm 0.69$ | $56.78 \pm 1.56$ | $67.53 \pm 0.51$ | $67.79 \pm 0.54$ | $59.36 \pm 0.93$ | $\mathbf{73.48 \pm 1.84}$ |
| | Type-III + 30% Asymmetric | $75.31 \pm 0.34$ | $74.62 \pm 1.71$ | $75.70 \pm 0.91$ | $76.09 \pm 0.10$ | $77.19 \pm 0.74$ | $\mathbf{77.54 \pm 0.70}$ |
| CIFAR-100 | Type-I + 30% Uniform | $48.86 \pm 0.56$ | $52.33 \pm 0.64$ | $52.90 \pm 0.53$ | $51.34 \pm 0.64$ | $45.66 \pm 1.60$ | $\mathbf{60.09 \pm 0.15}$ |
| | Type-I + 60% Uniform | $35.97 \pm 1.12$ | $27.17 \pm 1.66$ | $38.62 \pm 1.65$ | $37.57 \pm 0.43$ | $23.37 \pm 0.72$ | $\mathbf{51.68 \pm 0.10}$ |
| | Type-I + 30% Asymmetric | $45.85 \pm 0.93$ | $51.21 \pm 0.31$ | $52.69 \pm 1.14$ | $50.18 \pm 0.97$ | $52.04 \pm 0.15$ | $\mathbf{56.40 \pm 0.34}$ |
| | Type-II + 30% Uniform | $49.32 \pm 0.36$ | $51.99 \pm 0.75$ | $53.61 \pm 0.46$ | $50.58 \pm 0.25$ | $43.86 \pm 1.31$ | $\mathbf{60.01 \pm 0.63}$ |
| | Type-II + 60% Uniform | $35.16 \pm 0.05$ | $25.91 \pm 0.64$ | $39.58 \pm 3.13$ | $37.93 \pm 0.22$ | $23.05 \pm 0.99$ | $\mathbf{49.35 \pm 1.53}$ |
| | Type-II + 30% Asymmetric | $46.50 \pm 0.95$ | $51.07 \pm 1.44$ | $51.98 \pm 0.37$ | $49.46 \pm 0.23$ | $52.11 \pm 0.46$ | $\mathbf{61.43 \pm 0.33}$ |
| | Type-III + 30% Uniform | $48.94 \pm 0.61$ | $49.94 \pm 0.44$ | $52.07 \pm 0.35$ | $50.18 \pm 0.54$ | $42.79 \pm 1.78$ | $\mathbf{60.14 \pm 0.97}$ |
| | Type-III + 60% Uniform | $34.67 \pm 0.16$ | $22.89 \pm 0.75$ | $36.82 \pm 0.49$ | $37.65 \pm 1.42$ | $22.81 \pm 0.72$ | $\mathbf{50.73 \pm 2.16}$ |
| | Type-III + 30% Asymmetric | $45.70 \pm 0.12$ | $49.38 \pm 0.86$ | $50.87 \pm 1.12$ | $48.15 \pm 0.90$ | $50.31 \pm 0.39$ | $\mathbf{54.56 \pm 1.11}$ |

Table 3: The effect of $\theta_0$ on the performance. We use CIFAR-10 with 35% feature-dependent noise, and set $\beta = 0.1$.

| $\exp(\theta_0)$ | 0.2 | 0.3 | 0.4 | 0.5 |
|------------------|-----|-----|-----|-----|
| Type-I Noise | 83.33 | 83.04 | 82.66 | 82.94 |
| Type-II Noise | 81.84 | 81.18 | 81.09 | 81.24 |
| Type-III Noise | 81.79 | 81.75 | 81.98 | 82.06 |

Table 4: The effect of $\beta$ on the performance. We use CIFAR-10 with 35% feature-dependent noise, and set $\exp(\theta_0) = 0.3$.

| $\beta$ | 0.05 | 0.1 | 0.2 | 0.3 |
|---------|------|-----|-----|-----|
| Type-I Noise | 83.58 | 83.04 | 83.28 | 83.31 |
| Type-II Noise | 80.94 | 81.18 | 80.98 | 80.86 |
| Type-III Noise | 81.91 | 81.75 | 82.13 | 82.39 |

**Results on real-world noisy datasets.** To test the effectiveness of the proposed method under real-world label noise, we conduct experiments on the Clothing1M dataset (Xiao et al., 2015). This dataset contains 1 million clothing images obtained from online shopping websites with 14 categories. The labels in this dataset are quite noisy with an unknown underlying structure. This dataset provides 50*k*,

14*k* and 10*k* manually verified clean data for training, validation and testing, respectively. Following (Tanaka et al., 2018; Yi & Wu, 2019), in our experiment we discard the 50*k* clean training data and evaluate the classification accuracy on the 10*k* clean data. Also, following (Yi & Wu, 2019), we use a randomly sampled pseudo-balanced subset as the training set, which includes about 260*k* images. We set the batch size 32, learning rate 0.001, and adopt SGD optimizer and use ResNet-50 with weights pre-trained on ImageNet, as in (Tanaka et al., 2018; Yi & Wu, 2019).

We compare our method with the following baselines. (1) Standard; (2) Forward Correction (Patrini et al., 2017); (3) D2L (Ma et al., 2018); (4) JO (Tanaka et al., 2018); (5) PENCIL (Yi & Wu, 2019); (6) DY (Arazo et al., 2019); (7) GCE (Zhang & Sabuncu, 2018); (8) SL (Wang et al., 2019); (9) MLNT (Li et al., 2019); (10) LRT (Zheng et al., 2020). In Table 5 we observe that our method achieves the best performance, suggesting the applicability of our label correction strategy in real-world scenarios.

Apart from Clothing1M, we also test our method on another smaller dataset, Food-101N (Lee et al., 2018). Food-101N is a dataset for food classification, and consists of 310k training images collected from the web. The estimated label purity is 80%. Following (Lee et al., 2018), the classification accuracy is evaluated on the Food-101 (Bossard et al., 2014) testing set, which contains 25k images with curated annotations. We use ResNet-50 pre-trained on ImageNet. We train the network for 30 epochs with SGD optimizer. The batch size is 32 and the initial learning rate is 0.005, which is divided by 10 every 10 epochs. We also adopt simple data augmentation procedures, including random horizontal flip, and resizing the image with a short edge of 256 and then randomly cropping a 224x224 patch from the resized image. We repeat the experiments with 3 random trials and report the mean value and standard deviation. The results are shown in Table 6. Our method much improves upon the previous approaches.

Finally, we test our method on a recently proposed real-world dataset, ANIMAL-10N (Song et al., 2019). This dataset contains human-labeled online images for 10 animals with confusing appearance. The estimated label noise rate is 8%. There are 50,000 training and 5,000 testing images. Following (Song et al., 2019), we use VGG-19 with batch normalization. The SGD optimizer is employed. Also following (Song et al., 2019), we train the network for 100 epochs and use an initial learning rate of 0.1, which is divided by 5 at 50% and 75% of the total number of epochs. We repeat the experiments with 3 random trials and report the mean value and standard deviation. As is shown in Table 7, our method outperforms the existing baselines.

Table 5: Test accuracy (%) on Clothing1M.

| Method | Standard | Forward | D2L | JO | PENCIL | DY | GCE | SL | MLNT | LRT | **PLC (ours)** |
|---|---|---|---|---|---|---|---|---|---|---|---|
| Accuracy | 68.94 | 69.84 | 69.47 | 72.23 | 73.49 | 71.00 | 69.75 | 71.02 | 73.47 | 71.74 | **74.02** |

Table 6: Test accuracy (%) on Food-101N.

| Method | Accuracy |
|---|---|
| Standard | 81.67 |
| CleanNet (Lee et al., 2018) | 83.95 |
| **PLC (ours)** | **85.28 ± 0.04** |

Table 7: Test accuracy (%) on ANIMAL-10N.

| Method | Accuracy |
|---|---|
| Standard | 79.4 ± 0.14 |
| SELFIE (Song et al., 2019) | 81.8 ± 0.09 |
| **PLC (ours)** | **83.4 ± 0.43** |

## 5 CONCLUSION

We propose a novel family of feature-dependent label noise that is much more general than the traditional i.i.d. noise pattern. Building upon this noise assumption, we propose the first data-recalibrating method that is theoretically guaranteed to converge to a well-behaved classifier. On the synthetic datasets, we show that our method outperforms various baselines under different feature-dependent noise patterns subject to our assumption. Also, we test our method on different real-world noisy datasets and observe superior performances over existing approaches. The proposed noise family offers a new theoretical setting for the study of label noise.

**Acknowledgement.** The authors acknowledge support from US National Science Foundation (NSF) awards CRII-1755791, CCF-1910873, CCF-1855760. This effort was partially supported by the Intelligence Advanced Research Projects Agency (IARPA) under the contract W911NF20C0038. The content of this paper does not necessarily reflect the position or the policy of the Government, and no official endorsement should be inferred.

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

# A   APPENDIX

**Lemma 1** (One round purity improvement). *Suppose Assumption 1 is satisfied, and assume an $f$ such that there exists a pure $(e, f, \eta)$-level set with $3\epsilon \leq e < \frac{1}{2}$. Let $\tilde{\eta}_{new}(\boldsymbol{x}) = y_{f(\boldsymbol{x})}$ if $|f(\boldsymbol{x}) - 1/2| \geq e$ and $\tilde{\eta}(\boldsymbol{x})$ if $|f(\boldsymbol{x}) - 1/2| < e$, and assume $f_{new} = \arg\min_{h \in \mathcal{H}} E_{(\boldsymbol{x},\tilde{y}) \sim D(\boldsymbol{x}, \tilde{\eta}_{new}(\boldsymbol{x}))} Loss(h(\boldsymbol{x}), \tilde{y})$. Let $e_{new} = \min\{e | e > 0, \ L(e, \eta) \text{ is pure for } f_{new}\}$. Then $\frac{1}{2} - e_{new} \geq (1 + \frac{\varepsilon}{\alpha\ell})(\frac{1}{2} - e)$.*

**Proof**: We analyze the case where $\eta(\boldsymbol{x}) > \frac{1}{2}$. The analysis on the other side can be derived similarly. Due to the fact that there exists a level set $L(e, \eta)$ pure to $f$, we have $e \leq |\eta(\boldsymbol{x}) - \frac{1}{2}|$, $\forall \boldsymbol{x}$:

$$\mathbb{E}_{(\boldsymbol{z},\tilde{y}) \sim (D, \tilde{\eta}_{new}(\boldsymbol{z}))} \left[ \mathbb{1}_{\{\tilde{y}_{\boldsymbol{z}} \neq \eta^*(\boldsymbol{z})\}}(\boldsymbol{z}) \middle| \eta(\boldsymbol{z}) - \frac{1}{2} \middle| \geq \middle| \eta(\boldsymbol{x}) - \frac{1}{2} \middle| \right] = 0.$$

Now consider $\boldsymbol{x}$ where $e - \gamma \leq |\eta(\boldsymbol{x}) - \frac{1}{2}|$. Since the distribution $D$ is $(c_*, c^*)$-bounded, we have:

$$\mathbb{E}_{(\boldsymbol{z},\tilde{y}) \sim (D, \tilde{\eta}_{new}(\boldsymbol{z}))}[\mathbb{1}_{\{\tilde{y}_{\boldsymbol{z}} \neq \eta^*(\boldsymbol{z})\}}(\boldsymbol{z}) || \eta(\boldsymbol{z}) - \frac{1}{2} | \geq |\eta(\boldsymbol{x}) - \frac{1}{2}|]$$

$$= \mathbb{P}_{\boldsymbol{z}}[\tilde{y}_{\boldsymbol{z}} \neq \eta^*(\boldsymbol{z}) || \eta(\boldsymbol{z}) - \frac{1}{2}| \geq |\eta(\boldsymbol{x}) - \frac{1}{2}|]$$

$$= \frac{\mathbb{P}_{\boldsymbol{z}}[\tilde{y}_{\boldsymbol{z}} \neq \eta^*(\boldsymbol{z}), |\eta(\boldsymbol{z}) - \frac{1}{2}| \geq |\eta(\boldsymbol{x}) - \frac{1}{2}|]}{\mathbb{P}_{\boldsymbol{z}}[|\eta(\boldsymbol{z}) - \frac{1}{2}| \geq |\eta(\boldsymbol{x}) - \frac{1}{2}|]}$$

$$\leq \underbrace{\frac{\mathbb{P}_{\boldsymbol{z}}[\tilde{y}_{\boldsymbol{z}} \neq \eta^*(\boldsymbol{z}), \frac{1}{2} + e - \gamma \leq \eta(\boldsymbol{z}) \leq \frac{1}{2} + e]}{\mathbb{P}_{\boldsymbol{z}}[|\eta(\boldsymbol{z}) - \frac{1}{2}| \geq |\eta(\boldsymbol{x}) - \frac{1}{2}|]} + \frac{\mathbb{P}_{\boldsymbol{z}}[\tilde{y}_{\boldsymbol{z}} \neq \eta^*(\boldsymbol{z}), \frac{1}{2} - e \leq \eta(\boldsymbol{z}) \leq \frac{1}{2} - e + \gamma]}{\mathbb{P}_{\boldsymbol{z}}[|\eta(\boldsymbol{z}) - \frac{1}{2}| \geq |\eta(\boldsymbol{x}) - \frac{1}{2}|]}}_{\leq \frac{c^* \gamma}{c_*(\frac{1}{2} - e + \gamma)}}$$

$$+ \underbrace{\frac{\mathbb{P}_{\boldsymbol{z}}[\tilde{y}_{\boldsymbol{z}} \neq \eta^*(\boldsymbol{z}), \frac{1}{2} + e \leq \eta(\boldsymbol{z})]}{\mathbb{P}_{\boldsymbol{z}}[|\eta(\boldsymbol{z}) - \frac{1}{2}| \geq |\eta(\boldsymbol{x}) - \frac{1}{2}|]} + \frac{\mathbb{P}_{\boldsymbol{z}}[\tilde{y}_{\boldsymbol{z}} \neq \eta^*(\boldsymbol{z}), \eta(\boldsymbol{z}) \leq \frac{1}{2} - e]}{\mathbb{P}_{\boldsymbol{z}}[|\eta(\boldsymbol{z}) - \frac{1}{2}| \geq |\eta(\boldsymbol{x}) - \frac{1}{2}|]}}_{= 0}$$

$$= \frac{c^* \gamma}{c_*(\frac{1}{2} - e + \gamma)}.$$

If $\frac{\varepsilon}{\ell\alpha}(\frac{1}{2} - e) \leq \gamma \leq \frac{2\varepsilon}{\ell\alpha}(\frac{1}{2} - e)$, the impurity in super level set $\frac{1}{2} + e - \gamma$ is at most $\frac{2\epsilon}{\alpha}$. The level set $(\alpha, \varepsilon)$-consistency condition implies $|f_{new}(\boldsymbol{x}) - \eta(\boldsymbol{x})| \leq 3\epsilon$ for $\boldsymbol{x}$ s.t. $e - \gamma \leq |\eta(\boldsymbol{x}) - \frac{1}{2}|$. If $e \geq 3\epsilon$, $f_{new}(\boldsymbol{x})$ will give the same label as $\eta^*(\boldsymbol{x})$ and thus $(e - \gamma, \eta)$-level set becomes pure for $f$. Meanwhile, the choice of $\gamma$ ensures that $\frac{1}{2} - e_{new} \geq (1 + \frac{\varepsilon}{\ell\alpha})(\frac{1}{2} - e)$. $\qquad \square$

**Lemma 2** (Warm-up rounds). *Suppose for a given function $f_0$ there exists a level set $L(e_0, \eta)$ which is pure for $f_0$. Given $T_0 < 1/2$, after running Algorithm 1 for $m \geq \frac{\ell\alpha}{\varepsilon} \log(\frac{2T_0}{1 - 2e_0})$ rounds, there exists a level set $L(\frac{1}{2} - T_0, \eta)$ that is pure for $f_0$.*

**Proof**: The proof follows from the fact that each round of label flipping improves the purity by a factor of $(1 + \frac{\varepsilon}{\ell\alpha})$. To obtain an at least $T_0$ pure region, it suffices to repeat the flipping step for $m \geq \frac{\ell\alpha}{\varepsilon} \log(\frac{T_0}{1-2e_0})$ rounds. □

**Lemma 3.** *Suppose Assumption 1 is satisfied, and for a given function $f_0$ there exists a level set $L(e_0, \eta)$ which is pure for $f_0$. If one runs Algorithm 1 starting with $f_0$ and the initializations: (1) $T_0 < \frac{1}{2} - e_0$, (2) $m \geq \frac{\ell\alpha}{\varepsilon} \log(\frac{2T_0}{1-2e_0})$, (3) $N \geq m + \frac{1}{\beta} \log(\frac{1-6\varepsilon}{2T_0})$, (4) $T_{end} \leq \frac{1}{2} - 3\epsilon$ and (5) $\frac{\varepsilon}{\alpha\ell} \leq \beta \leq \frac{2\varepsilon}{\alpha\ell}$, then we have $\mathbb{P}_{x \sim D}[y_{f_{final}(x)} = \eta^*(x)] \geq 1 - 3c^*\epsilon$.*

**Proof**: The proof can be done by combining Lemma 1 and Lemma 2. In the first $m$ iterations, by Lemma 2, we can guarantee a level set $(\frac{1}{2} - T_0, \eta)$ pure to $f$. In the rest of the iterations we ensure the level set $|\eta(x) - \frac{1}{2}| \geq \frac{1}{2} - T$ is pure. We increase $T$ by a reasonable factor of $\beta$ to avoid incurring too many corrupted labels while ensuring enough progress in label purification, i.e., $\frac{\varepsilon}{\alpha\ell} \leq \beta \leq \frac{2\varepsilon}{\alpha\ell}$, such that in the level set $|\eta(x) - \frac{1}{2}| \geq \frac{1}{2} - T$ we have $|f(x) - \eta(x)| \leq 3\varepsilon$. This condition ensures the correctness of flipping when $T \leq \frac{1}{2} - 3\varepsilon$. The purity cannot be improved once $T \geq \frac{1}{2} - 3\varepsilon = T_{end}$ since there is no guarantee that $f(x)$ has consistent label with $\eta(x)$ when $|\eta(x) - \frac{1}{2}| < 3\varepsilon$ and $|\eta(x) - f(x)| \leq 3\varepsilon$. By $(c_*, c^*)$-bounded assumption on $D$, its mass of impure $3\varepsilon$ level set region is at most $3c^*\epsilon$. □

**Theorem 1.** *Under assumption 1, for any noise $\tau$ which is PMD with margin $t_0$, define $e_0 = max(t_0, \frac{\alpha+\varepsilon}{1+2\alpha})$. Then for the output of Algorithm 1 with $f$ as above and with the following initializations: (1) $T_0 < \frac{1}{2} - e_0$, (2) $m \geq \frac{\ell\alpha}{\varepsilon} \log(\frac{2T_0}{1-2e_0})$, (3) $N \geq m + \frac{1}{\beta} \log(\frac{T_0}{3\varepsilon})$, (4) $T_{end} \leq 3\epsilon$ and (5) $\frac{\varepsilon}{\alpha\ell} \leq \beta \leq \frac{2\varepsilon}{\alpha\ell}$, we have:*

$$\mathbb{P}_{x \sim D}[y_{f_{final}(x)} = \eta^*(x)] \geq 1 - 3c^*\epsilon.$$

**Proof**: The proof is based on Lemma 3 plus a verification of the existence of $f_0$ for which there exists a pure $(e_0, f, \eta)$-level set. Let:

$$\tau(x) = \begin{cases} \tau_{10}(x) & \eta(x) \geq 1/2 \\ \tau_{01}(x) & \eta(x) < 1/2 \end{cases}.$$

In the level set $|\eta(x) - \frac{1}{2}| \geq e_0$, $\mathbb{P}_z[\tilde{y}_z \neq \eta^*(z)| |\eta(z) - \frac{1}{2}| \geq e_0] \leq \frac{1}{2} - e_0 + \tau(z)$. By level set $(\alpha, \varepsilon)$-consistency, it suffices to satisfy $\alpha(\frac{1}{2} - e_0 + \tau) + \varepsilon \leq e_0$ to ensure that $f(x)$ has the same prediction with $\eta(x)$ when $|\eta(x) - \frac{1}{2}| \geq e_0$. By polynomial level set diminishing noise, we have $\tau(x) \leq \frac{1}{2} - e_0$ if $e_0 > t_0$, and thus by choosing $e_0 = max(t_0, \frac{\alpha+\varepsilon}{1+2\alpha})$ one can ensure that initial $f_0$ has a pure $(e_0, f_0, \eta)$-level set. The rest of the proof follows from Lemma 3. □

