# OpenReview forum: "Learning with Feature-Dependent Label Noise: A Progressive Approach"
_ICLR.cc/2021/Conference — ICLR 2021 Spotlight_

### Official Review · AnonReviewer3 · 2020-10-19
**A good practical method and theoretical contribution for learning from feature dependent label noise**

**Rating:** 8
**Confidence:** 4

**Review:**

The paper presents a learning method for the scenario of feature dependent label noise. A framework where label noise diminishes away from the decision boundary is established and a relabeling strategy based on this by relabeling highly confident points is proposed.  The method is a straight-forward adaptive method which the authors both theoretically and empirically explore in detail.

# Pros

- The approach is simple and easy to implement on top of existing methods
- The authors support their simple method with some very nice theoretical results
- Good empirical evaluation showing competitive performance to other methods
- The paper is clearly written and easy to read
- Citations place the work well among existing literature

# Cons

- The authors mention Menon et al. in their "related works" section, but dismiss it immediately as "...it does not recalibrate individual data based on their contexts, and thus are not as effective as other deep-learning -based methods in practice." A citation here is needed showing this point, or further experiments.
- The related works section is very short and is essentially a list of other approaches. A more thorough discussion would help readers.
- A selection of baseline methods are chosen for comparison to the proposed approach in the empirical evaluation, but the particular choices aren't discussed in any detail. Some of the methods were mentioned in the related works section, but some are not (why not?), and neither section explains some of the choices.

---

> ### Author Response · Authors · 2020-11-18
> **Literature part is expanded and synthetic experiment is added**
>
> **Q1:** The authors mention Menon et al. in their "related works" section, but dismiss it immediately as "...it does not recalibrate individual data based on their contexts, and thus are not as effective as other deep-learning-based methods in practice." A citation here is needed showing this point, or further experiments.
>
> **A:** Thank you for pointing this out. Let us revise our statement as follows: (Menon et al. 2018) provides an elegant theoretical framework, showing that losses fulfilling certain conditions naturally resist instance-dependent noise. The method can achieve even better theoretical properties (i.e., Bayes-consistency) with stronger assumption on the clean posterior probability $\eta$. In practice, this method has not been extended to deep neural networks. We empirically compare this method with ours. We follow the experimental setting of Menon et al. [1] and inject different level BCN noise into MNIST data set. We observed that PCL outperforms the Isotron method by Menon et al. [1]. The results are as follows.
>
> | MNIST  | Menon’s Isotron  |  PCL  |
> |:---:|:---:|:---:|
> |30%  | 95.65 ± 0.00 | 97.55 ± 1.07 |
> |40%  | 87.68 ± 0.00 | 91.98 ± 2.66 |
> |45%  | 80.88 ± 0.03 | 86.69 ± 3.48 |
> ||||
>
> We have updated the discussion of this method in the related work section accordingly.
>
> **Q2:** The related works section is very short and is essentially a list of other approaches. A more thorough discussion would help readers.
>
> **A:** We added more discussion of the literature in the related work section.
>
> **Q3:** A selection of baseline methods are chosen for comparison to the proposed approach in the empirical evaluation, but the particular choices aren't discussed in any detail. Some of the methods were mentioned in the related works section, but some are not (why not?), and neither section explains some of the choices.
>
> **A:** We select a collection of recent methods representing different strategies. Co-teaching+ is a data-selection method: it selects the clean data by training two networks simultaneously, and then updates the networks on the selected clean data. GCE and SL aim to combat the label noise with noise-tolerant losses. LRT not only detects the noisy labels but also corrects them. This way more training data are utilized, compared with data-selection methods. We think these methods are reasonably representative of the recent developments in learning with label noise.
>
> Thanks for the suggestion regarding the related work. We add discussions of all the selected baselines in the related work section. Also, we will provide more details and discussions on these methods.
>
> [1] Learning from Binary Labels with Instance-Dependent Corruption. Aditya Krishna Menon, Brendan van Rooyen, Nagarajan Natarajan. arXiv preprint arXiv:1605.00751

---

### Official Review · AnonReviewer1 · 2020-10-26
**The authors introduce an iterative approach to learning from data with noisy labels under realistic circumstances (rather than iid assumptions).**

**Rating:** 7
**Confidence:** 3

**Review:**

The work appears to original, and it deals with an important topic (ie, how to deal with noisy labels). The paper is well written and reasonably easy to follow. Figure 1 is extremely helpful in providing an intuitive explanation that helps understanding the rest of the paper.

Even though the empirical validation is reasonably thorough for a paper that also includes a theoretical analysis of the novel approach, it is here that the authors could improve the paper the most. The main weakness of this section is that even though the results on the synthetic datasets are strong, the ones on the Clothing1M domain are significantly less so, with two competing approaches (PENCIL  & MLTN) within 0.53% - 0.55% from the PCL result. Ideally, the paper should include another 2-3 real-world domains in this evaluation. At the very least, they could add to the evaluation the CUB-200 dataset from [Yi & Wu, 2019], on which PENCIL was evaluated for robustness on less noisy data.  Adding two more synthetic domains such as MNIST and ModelNet40, as done in [Zheng et al, 2020], is another possibility - but far less appealing.

---

> ### Author Response · Authors · 2020-11-18
> **Additional real-world datasets are added**
>
> **Q:**  Ideally, the paper should include another 2-3 real-world domains in this evaluation. At the very least, they could add to the evaluation the CUB-200 dataset from [Yi & Wu, 2019]
>
> **A:**  Thanks for the suggestion. We conduct experiments on three additional real-world datasets, including CUB-200.
>
> (1) Food-101N [1], which is a dataset for food classification. It consists of 310k training images collected from the web. The estimated label purity is 80%. Following [1], the classification accuracy is evaluated on the Food-101 [2] test set, which contains 25k images with curated annotations. We use ResNet-50 pretrained on ImageNet. We train the network for 30 epochs with SGD optimizer. The batch size is 32 and the initial learning rate is 0.005, which is divided by 10 every 10 epochs. We also adopt simple data augmentation procedures, including random horizontal fiip, and resizing the image with a short edge of 256 and then randomly cropping a 224x224 patch from the resized image. We repeat the experiments with 3 random trials and report the mean value and standard deviation. The results are as follows:
>
> |||
> |:---|---:|
> |Standard         | 81.67|
> |CleanNet [1]   | 83.95|
> |PCL (ours)      | 85.28±0.04|
> |||
>
> Our method achieves better performance than CleanNet. Note that we do not use the extra 55k manually cleaned training data, which are utilized by CleanNet for noise detection.
>
> (2) ANIMAL-10N [3], which is a real-world noisy dataset consisting of human-labeled online images for 10 animals with confusing appearance. The estimated label noise rate is 8%. There are 50,000 training and 5,000 testing images. Following [3], we use VGG-19 with batch normalization. The SGD optimizer is employed. Following [3], we train the network for 100 epochs and use an initial learning rate of 0.1, which is divided by 5 at 50% and 75% of the total number of epochs. We repeat the experiments with 3 random trials and report the mean value and standard deviation. The results are as follows:
>
> |||
> |:---|---:|
> |Standard         | 79.4±0.14|
> |SELFIE [3]      | 81.8±0.09|
> |PCL (ours)      | 83.4±0.43|
> |||
>
> We observe that our method achieves better performance than SELFIE.
>
> (3) Following the suggestion of AnonReviewer1, we conduct experiments on the CUB-200 [4] which is for fine-grained bird classification. The labels in this dataset are considered clean. We use ResNet-50 pretrained on ImageNet. Simple data augmentation procedures are also applied, including horizontal random flip, and resizing the image to 256x256 and then randomly cropping a 224x224 patch from the resized image. We use SGD optimizer and batch size of 16. The initial learning rate is 0.002, which is divided by 10 after 100, 120 and 140 epochs. The total number of training epochs is 160.
>
> The standard method (using cross entropy loss alone) achieves classification accuracy 79.91±0.04. Our PCL method has accuracy 79.99±0.04. The performance of our method does not drop for a dataset without any noise.
>
> [1] Kuang-Huei Lee, Xiaodong He, Lei Zhang, and Linjun Yang. Cleannet: Transfer learning for scalable image classifier training with label noise. In CVPR, 2018.
>
> [2] Lukas Bossard, Matthieu Guillaumin, and Luc Van Gool. Food-101– mining discriminative components with random forests. In ECCV, 2014.
>
> [3] Hwanjun Song, Minseok Kim, and Jae-Gil Lee. SELFIE: Refurbishing Unclean Samples for Robust Deep Learning. In ICML, 2019.
>
> [4] P. Welinder, S. Branson, T. Mita, C. Wah, F. Schroff, S. Belongie, and P. Perona. Caltech-UCSD Birds 200. Technical Report CNS-TR-2010-001, California Institute of Technology, 2010.
>
> [5] Kun Yi and Jianxin Wu. Probabilistic end-to-end noise correction for learning with noisy labels. In CVPR, 2019.

---

### Official Review · AnonReviewer4 · 2020-10-26
**Simple and effective label de-noising algorithm**

**Rating:** 8
**Confidence:** 3

**Review:**

**Summary of paper**

The authors introduce a data-relabeling method that they claim is the first that both allows for data-dependent noise and is theoretically guaranteed to converge to an optimal model.

The authors introduce a novel family of label noise, called Polynomial Margin Diminishing (PMD), which defines a polynomially-decreasing upper bound on the label noise as the true label probability is above some threshold and as it approaches 1. Below the threshold (when the true label probability is closer to 0.5), the noise is unbounded.

The authors introduce an algorithm, "Progressive Label Correction", which iteratively flips the training labels of examples for which the model's confidence is above a threshold. The threshold decreases over time, so that only the most confident examples are flipped at first, and then the less-confident examples are flipped later.

The authors prove (Theorem 1) that under the assumption of PMD label noise distribution, as well as Assumption 1 concerning the flexibility of the hypothesis class and the continuity of the true label's conditional distribution, then their Progressive Label Correction algorithm asymptotically approaches a set of corrected labels that match the true (de-noised) labels with high probability.

The authors experimentally show that their algorithm consistently outperforms 5 alternatives on CIFAR-10 and CIFAR-100 datasets with various types of synthetic noise, both feature-dependent and hybrid feature-depedent/indepdendent. Finally, they show their algorithm outperforms 10 alternatives on a real-world dataset (Clothing1M) with unknown noise.

**Conclusions**

Quality: Overall I like this paper. It's a pretty simple algorithm to implement, and it seems to be quite effective in practice and have nice theoretical properties.

Clarity: The paper is structured very well. It is easy to follow the narrative and high-level ideas. There are a few minor typos (see below for some examples). The theory is relatively easy to follow.

Originality: I'm not familiar with the related work, but this appears to be original/novel from my limited perspective.

Significance: This algorithm can potentially be used to improve test accuracy on any supervised learning task, so the intended audience is quite large. The improvement on both synthetic and real data seems quite large. It's hard to tell whether the results were cherry-picked at all, but they are impressive.

**Minor comments**

Figre 1 caption has a typo: "Red dots are the data that remain incorrect. that remain un-corrected and are closer to the
decision boundary."

In Section 2.1, "illustrate the upperbound (red curve)", but the figure actually shows an orange curve (not red) for the upper bound.

Figure 2 caption, "closed to 0 or 1" should be "close to 0 or 1". Also, "a equal probability" should be "an equal probability".

---

> ### Author Response · Authors · 2020-11-18
> **We've fixed typos in the paper**
>
> **Q:** Typos in the paper
>
> **A:** Thank you so much for the positive feedback. We have fixed the typos according to your comments. Please refer to the paper for the update.

---

### Official Review · AnonReviewer2 · 2020-10-27
**This paper proposes a progressive label correction algorithm by correcting labels and refine the model iteratively.**

**Rating:** 7
**Confidence:** 4

**Review:**

Comments:
Label noise is very frequently in many real world applications. However, the noise can be with different distributions. If we build the learning model under a certain distribution, it is difficult to capture the discriminative information. In this paper, without assuming that the noise is a certain distribution, the proposed method can handle the general noise, and it mainly target a new family of feature-dependent label noise, which is much more general than commonly used i.i.d. label noise and encompasses a broad spectrum of noise patterns. The experimental results show that the proposed method is promising. Meanwhile, the theoretical analysis of the proposed method is well inferred.

Strong points:
[1] The theoretical foundation of the proposed method is strong.
[2] The experimental results of the proposed method are promising.
Weak points:
[1] Some details about the experiments are not clear, such as the experimental settings of the compared methods.
[2] It is better to show the connection between the polynomial margin diminishing noise and the other noises.

Accept reason:
[1] The paper has shown a promising performance than several state-of-the-art methods. The noise assumption is more general than the traditional types. Hence, the paper may provide a novel way to deal with noise labels.

Feedbacks:
[1] I found that the step size \beta has an influence on the threshold \theta, and how to set it. It is necessary to show the details about \beta, which has directly influence on the results.
[2] The noise in the experiments is more than 30%. Whether the proposed method is suitable to the high-level noise. It is better to show the results without any noise.
[3] Since the polynomial margin diminishing noise is general, whether the polynomial margin diminishing noise can represent any noise functions in theoretical.
[4] The details of all the compared method may need to be provided.

---

> ### Author Response · Authors · 2020-11-18
> **Provide more experiments, ablation studies and discussions**
>
> **Q1:** It is better to show the connection between the polynomial margin diminishing noise and the other noises.
>
> **A:** In Section 2.1, we listed several existing types of noise: uniform and BCN. In most of the feature domain, our PMD noise is very flexible and generalizes these known noise types. The only exception is the region where $\eta$ is highly confident (almost 0 or almost 1). There our PMD noise requires a nontrivial noise upper bound.
>
>
> **Q2:**  It is necessary to show the details about $\beta$, which has a direct influence on the results.
>
> **A:** On CIFAR-10, the effect of the step size $\beta$ on the model performance is shown below (we set $\exp(\theta)$ to 0.3):
>
> | $\beta$ | 0.05 | 0.1| 0.2 | 0.3 |
> |:----:|:----:|:----:|:----:|:----:|
> |Type-I   noise  | 83.58 | 83.04 | 83.28 | 83.31|
> |Type-II  noise  | 80.94 | 81.18 | 80.98 | 80.86|
> |Type-III noise  | 81.91 | 81.75 | 82.13 | 82.39|
> ||
>
> We observe that our method is not sensitive to $\beta$.
>
>
> **Q3:** It is better to show the results without any noise.
>
> **A:**  Thanks for the suggestion. We applied our method to CIFAR-10 and CIFAR-100 without any label noise. It achieved similar performance as the standard method. The results are as follows:
>
> ||||
> |:----:|:----:|:----:|
> |CIFAR-10   | Standard    | 93.29±0.59|
> |CIFAR-10   | PCL (ours) | 93.60±0.02|
> |CIFAR-100 | Standard    | 74.30±0.39|
> |CIFAR-100 | PCL (ours) | 74.25±0.09|
> ||
>
> **Q4:** Since the polynomial margin diminishing noise is general, whether the polynomial margin diminishing noise can represent any noise functions theoretically.
>
> **A:** As discussed in Q1, the PMD noise can represent arbitrarily noise function in most of the feature domain. The only exception is the area where $\eta$ is highly confident (almost 0 or almost 1).
>
> **Q5:** Some details about the experiments are not clear, such as the experimental settings of the compared methods.
>
> **A:** Regarding common hyperparameters, we use the same setting for all methods: batch size = 128, total epoch = 180, and SGD optimizer with initial learning rate = 0.01. Some other model-specific hyperparameters are tuned based on the recommendation of the papers and validation performance.

---

### Public Comment · ~Ehsan_Amid1 · 2020-11-10
**Please consider referencing/comparing to these more recent works**

I would like to point out that our work (Amid et al. 2019a) extends the Generalized CE loss (Zhang and Sabuncu 2018) by introducing two temperatures t1 and t2 which recovers GCE when t1 = q and t2 = 1. Our more recent work, called the bi-tempered loss (Amid et al. 2019b) extends these methods by introducing a proper (unbiased) generalization of the CE loss and is shown to be extremely effective in reducing the effect of noisy examples. Please consider referencing/comparing to these papers.

(Amid et al. 2019a) Amid et al. "Two-temperature logistic regression based on the Tsallis divergence." In The 22nd International Conference on Artificial Intelligence and Statistics (AISTATS), 2019.

(Amid et al. 2019b) Amid et al. "Robust bi-tempered logistic loss based on Bregman divergences." In Advances in Neural Information Processing Systems (NeurIPS), 2019.

---

> ### Author Response · Authors · 2020-11-18
> **Thanks for your suggestion**
>
> **Q:** Please consider referencing/comparing to these papers.
>
> **A:**  Thanks for the suggestion. We have added discussion of these works. Please see the related work section for the update.

---

### Author Response · Authors · 2020-11-18
**Thank you for your constructive and helpful comments.**

We thank reviewers for constructive and helpful comments. We are glad all reviewers are thinking positively about this paper. Per their suggestions, we have added more experimental results on real-world dataset and ablation studies. We have also improved the presentation of our manuscript. Below we will address the concerns of each reviewer one-by-one.

---

### Comment · ~Pengfei_Chen1 · 2021-01-15
**A minor issue and a related work that provides rigorous motivations for learning with feature-dependent noise.**

Thanks for the excellent work on learning with instance(feature)-dependent noise (IDN)! I really appreciate the modeling on the noise and the theoretical guarantee on the performance of progressive label correction. Learning with IDN is challenging while deriving a theoretical guarantee for IDN is even more challenging since the noise is dependent on individual instance, unlike class-conditional noise (CCN) for which the noise can be modeled by class-dependent noise transition probabilities (namely the noise transition matrix).

In this paper, the author defines a natural feature-dependent noise and derive the theoretical guarantee based on two assumptions: Definition 2, which somewhat guarantees the accuracy on confident samples; Definition 3, which characterizes the imbalance of data distribution. For example, the data distribution could not be too imbalanced such that most samples are close to the decision boundary. This work is an interesting and important contribution!

There is a minor issue in a claim in Section 4: **this noise will hurt the network’s performance the most**. Strictly speaking, IDN is confusing and is difficult to mitigate, but in terms of test performance, CCN hurts more. Overfitting to random noise degenerates the generalization most, as demonstrated in the work [1] (Section 3.2 and Figure 4). Still, IDN is more challenging to mitigate since it is easier to overfit. Moreover, [1] provides **rigorous motivations for going beyond CCN and studying IDN. Notably, it is shown that the noise in Clothing1M is impossible to be feature *in*dependent, with a probability lower than $10^{-21250}$** (Section 2.2 and Theorem 1).

[1] Beyond Class-Conditional Assumption: A Primary Attempt to Combat Instance-Dependent Label Noise. arxiv.org/abs/2012.05458. (Accepted by AAAI 2021).

---

> ### Author Response · Authors · 2021-02-01
> **Response to the comment**
>
> Dear Pengfei,
>
> Thank you very much for your comment on our work. We appreciate the positive feedback. We have read the referred paper [1] and will cite it in our camera-ready version.
>
> We would like to clarify that the IDN in [1] is very different and indeed much more restrictive than the diminishing noise proposed in our paper. __Thus, the claims in [1] regarding IDN does not apply to our noise model.__  In particular, the IDN in [1] requires the noise to be positively correlated with the confidence of the Bayesian/clean classifier. Whereas in our paper, we only require the noise to be upper bounded in some high confidence region and allow it to be arbitrary everywhere else. See Figure 2 for a demonstration.
>
> Best,
>
> Authors

---

### Decision · Program_Chairs · 2021-01-07
**Final Decision**

**Decision:**

Accept (Spotlight)

**Comment:**

This paper studies the problem of learning from data that have been corrupted by label noise. The authors define a natural data-dependent noise condition, that allows the noise rate to be large close to the decision boundary, and provide a simple iterative method that eventually converges to the Bayes optimal classifier. The method is evaluated on both synthetic a real datasets. There was a consensus among the reviewers that this is an interesting contribution and I propose acceptance.